# Barriers to utilisation of oral rehydration solution and zinc in managing diarrhoea among under-5 children in Oyo State, Nigeria

Adeniyi Francis Fagbamigbe [iD] ,[1,2] Joseph Juma,[2] John Kariuki[2]

[1]Department of Epidemiology and Medical Statistics, Faculty of Public Health, College of Medicine, University of Ibadan, Ibadan, Nigeria
[2]Department of Epidemiology and Biostatistics, School of Public Health, Mount Kenya University, Nairobi, Kenya

**Correspondence to**
Dr Adeniyi Francis Fagbamigbe; franstel74@yahoo.com

### ABSTRACT

Despite its effectiveness, oral rehydration solution (ORS) and zinc use for managing diarrhoea among under-5 children (U5C) is low in Nigeria. We assessed the barriers to utilisation and sources of ORS/zinc in Oyo State, Nigeria. A cross-sectional mixed-methods design was adopted. Of the 1154 mothers in the quantitative study, only 71 (6.2%) reported recent U5C diarrhoea, of which 41 used ORS/zinc. Eleven of these 41 obtained ORS/zinc from private chemists, and six from government hospitals. Topmost barriers to utilisation of ORS/zinc are unavailability, unaffordability and poor awareness. Stakeholders should intensify efforts to sensitise women, and improve the availability and affordability of ORS and zinc therapy.

Nigeria has the highest burden of diarrhoea in sub-Saharan Africa. Compared with a global average of 70.6, the mortality attributed to diarrhoea in Nigeria among under-5 children (U5C) in 2016 is 327.3/100 000 diarrhoea cases.[1] WHO and UNICEF recommended that any child with diarrhoea symptoms must be given oral rehydration therapy (ORT) and paediatric zinc sulfate dispersible tablets (zinc) within 24 hours.[2 3] The oral rehydration solution (ORS) is the most common ORT. ORS utilisation is low in Nigeria.[4] Literature is replete with factors associated with ORS/zinc use in managing diarrhoea among U5C.[4] However, there is a paucity of data on barriers to utilisation of ORS/zinc in managing diarrhoea among U5C. We identified the barriers to utilisation of ORS/zinc in managing diarrhoea among U5C in Oyo State, Nigeria, and also assessed the outlets/facilities where these commodities were obtained.

A cross-sectional mixed-methods population-based study was conducted using both quantitative and qualitative methods in Oyo State, Nigeria using a case study approach where health programmers and paediatricians were actively engaged in the study planning and implementation. A total of 1154 mothers/guardians selected using three-stage cluster sampling (wards/communities/households) participated in the quantitative arm while purposively sampled one government senior staff member; two paediatricians had key informant interviews and two focus group discussions (FGDs) among nine mothers/guardians each in the qualitative study.

Only 71 (6.2%) reported recent diarrhoea episodes among their children. A total of 41 (57.7%) of these 71 mothers used ORS/zinc, lower than 34% across Nigeria.[4] The utilisation varied by respondents' characteristics. Among FGD participants, main barriers are unavailability, unaffordability, poor awareness and spousal approval. Similar patterns were found in the quantitative arm (table 1). All the key informants stated that ORS/zinc was available free of charge in public hospitals, contrary to unavailability reported by FGD participants. Some mothers/guardians did not know where ORS/zinc can be obtained. Of the 41 with recent U5C diarrhoea who used ORS/zinc in the quantitative study, 11 obtained the commodity from private chemists, and 6 each from government hospitals and government health centres (table 2).

The most common stated barriers to the use of ORS/zinc in the management of diarrhoea were availability, affordability and awareness. These are corroborated by existing literature.[3 5 6] ORS/zinc was obtained at hospitals, chemists, pharmacies and markets. To improve the use of ORS/zinc, sensitisation of mothers/guardians who have poor access to health facilities through radio jingles and programmes and distribution of free ORS/zinc should be

**Table 1** Reasons for not using ORS/zinc

| Reasons | Used recently* (n=41) | Didn't use recently* (n=30) | All* (n=71) |
|---|---|---|---|
| Unavailability | 25 | 22 | 47 |
| Unaffordability | 36 | 20 | 56 |
| Unawareness | 0 | 6 | 6 |
| Spousal approval | 0 | 16 | 16 |
| Others | 0 | 2 | 2 |
| None | 0 | 2 | 1 |
| Total | 41 | **30** | **71** |

*Multiple responses.
ORS, oral rehydration solution.

intensified.[5][6] There is a need to design community-level behaviour change components to enhance awareness, elimination and management of diarrhoea. There are needs for hospital management to ensure that ORS/zinc stock-outs in public hospitals are eliminated. Finally, further research is necessary to validate key informants' claim that the commodities are

**Table 2** Distribution of places where ORS/zinc was obtained

| Place | n |
|---|---|
| Government hospital | 6 |
| Government health centre | 6 |
| Public mobile clinic | 2 |
| Other public sectors | 1 |
| Private hospital, clinic | 11 |
| Pharmacy | 7 |
| Private chemist/ PMS | 5 |
| Itinerant drug seller | 1 |
| Others | 2 |
| Any | 41 |
| None | 30 |
| Total | 71 |

ORS, oral rehydration solution; PMS, Patient Medicine Sellers.

available free of charge, as it contradicts the mothers/guardians' stand.

**Acknowledgements** The authors acknowledge the study participants: mothers/guardians, under-5 children, paediatricians, staff of the Ministry of Health of Oyo State, Nigeria, and staff and students of Mount Kenya University, Nairobi.

**Contributors** AFF contributed to conception and study design and conducted the statistical analysis and interpretation of results and the drafting of the manuscript. AFF, JJ and JK reviewed the statistical analysis and revised the manuscript draft. All authors read and approved the final manuscript.

**Funding** The authors have not declared a specific grant for this research from any funding agency in the public, commercial or not-for-profit sectors.

**Competing interests** None declared.

**Patient and public involvement statement** The patients and the public were not involved in the study design but were involved in the study implementation and dissemination of findings.

**Patient consent for publication** Obtained.

**Ethics approval** This study involves human participants and ethical approval was obtained from the Oyo State Ministry of Health with approval number AD/13/479/1696. Participants gave written informed consent to participate in the study before taking part.

**Provenance and peer review** Not commissioned; externally peer reviewed.

**ORCID iD**
Adeniyi Francis Fagbamigbe http://orcid.org/0000-0001-9184-8258

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
