## [Reviewer comments · BMJ Paediatrics Open]

ARTICLE DETAILS

TITLE (PROVISIONAL)	Barriers to utilization and sources of oral rehydration salt and zinc in managing diarrhoea among under-five children in Oyo State, Nigeria: A clarion call
AUTHORS	Fagbamigbe, Adeniyi Francis Juma, Joseph Kariuki, John

VERSION 1 – REVIEW

REVIEWER	Reviewer name: Ms. Ronel Sewpaul Institution and Country: HSRC Human and Social Capabilities Division, South Africa Competing interests: None
REVIEW RETURNED	03-Mar-2022

GENERAL COMMENTS	- line 17: "Despite its effectiveness, Oral Rehydration Salt (ORS) and Zinc use for managing diarrhoea among under-five children (U5C) is low in Nigeria." - the abstract introduction says that ORS/zinc use is low, but there is no literature mentioning this in the paper's Introduction section. Is this already known fact before doing this study or is this study investigating the extent of use? Please include literature from other studies showing that ORS/zinc use is low in Nigeria or else remove this from the intro in the abstract. - line 41: "A total of 1,154 mothers/guardians selected using three-stage cluster sampling procedure participated in the quantitative arm while purposively sampled one government senior 43 personnel, two paediatricians had KIIs and two FGDs with 9 mothers/guardians each." - Sentences could be made clearer, please improve grammar and provide more detail on the sampling procedure i.e. the 3 stages in the cluster sampling for the 1154 guardians, how the selection of each of government personnel, paediatricians and mothers/guardians for the qualitative component was done and specify what KII stands for. - line 50: "Of the 41 with recent U5C diarrhoea in the quantitative study, 11 obtained.." - should this not read "Of the 41 with recent U5C diarrhoea who used ORS/zinc in the quantitative study, 11 obtained..?" - line 51 - what is PMS the abbreviation for - please put the full words at first mention and the abbreviation in brackets. - line 51: improve grammar for readability: perhaps add from i.e. "6 each from a government hospital and a government health centre" - general: the finding that ORS/zinc is in fact available and free of charge at hospitals but many mothers reported unavailability should be emphasized and the reasons why mothers reported unavailability should be emphasized and discussed. lines 57-63 touch on this a little. general: of a population based study, the sample of 71 mothers reporting recent diarrhoea is quite small. The authors should comment on the relatively small sample size which is perhaps underpowered for assessing barriers to ORS. For example, was this sample of 1154 part of a bigger study that aimed to do.... and this study conducted a secondary analysis of the sample of 71 to assess patterns of barriers to ORS.
---

REVIEWER	Reviewer name: Dr. Akanni Ibukun Ibukun Institution and Country: Obafemi Awolowo University, Nigeria Competing interests: None
REVIEW RETURNED	29-Mar-2022

GENERAL COMMENTS	line 44 "Oly" should read only. How valid is the view of key informants from the hospital on availability of the product? Maybe, there are other ways either through mystery clients or physical confirmation of the product. Promoting awareness and BCC as recommended is very appropriate. However, what channels will be more suitable to the targeted groups?
---

VERSION 1 – AUTHOR RESPONSE

The Editor
Associate Editor,
BMJ Paediatrics Open

Re: bmjpo-2022-001450 - "Barriers to utilization and sources of oral rehydration salt and zinc in managing diarrhoea among under-five children in Oyo State, Nigeria: A clarion call"

We appreciate the reviews and the comments from our respected reviewers. Kindly find below point by point to each of the comments raised

Editors Comments

Many thanks for your submission to BMJ Paediatrics Open. We require some revisions to the letter prior to publication. Please see the comments from our reviewers, along with my comments below. We look forward to your resubmission.

Thank you

1. Abstract (line 24): "Stakeholders should intensify efforts to sensitize women, improve availability and affordability." should have "to Oral rehydration and zinc therapy
Thank you. This has been corrected
2. Line 30 (pg 4): Remove "whereas" at start of sentence
Thank you. This has been corrected
3. Line 33: ORS stands for Oral rehydration solution not salt
Thank you. This has been corrected
4. Lines 40-41 "whereby stakeholders were actively engaged" - who were the stakeholders and how were they engaged?
Thank you. We have provided this
5. Lines 41-43 - as with the review, this methodology is not clear, the acronyms used have not been defined.
Thank you. This has been corrected
6. What were the 3 stages, why were 3 stages needed?
Thank you. The stages were i. ward selection stage ii. Community selection stage and iii. Household selection stage. They were to ensure the representativeness of the study population
7. Is the second part of this -"while purposively sampled one government senior personnel, two paediatricians had KIIs and two FGDs with 9 mothers/guardians each" - do these indicate the people who were recruited to the qualitative arm of the study? if so please specify/clarify this, if not this needs further explanation
8. Yes, these were the participants in the qualitative study. It has been indicated
9. Pg5 Line 60 - "Sensitisation of women" should refer to "caregivers" or guardians, rather than just women, (you have used guardians in previous sentences in the letter)

Thank you. This has been corrected

10. It would be useful to compare the proportion of caregivers you found had accessed ORS/Zinc within your study to any other statistics surrounding this - does 57% fit with other studies around this topic? If not then compare your data to this other data
Thank you, we have added the comparative data

11. Reviewer: 1

Ms. Ronel Sewpaul, HSRC

12. line 17: "Despite its effectiveness, Oral Rehydration Salt (ORS) and Zinc use for managing diarrhoea among under-five children (U5C) is low in Nigeria." - the abstract introduction says that ORS/zinc use is low, but there is no literature mentioning this in the paper's Introduction section. Is this already known fact before doing this study or is this study investigating the extent of use? Please include literature from other studies showing that ORS/zinc use is low in Nigeria or else remove this from the intro in the abstract.

We agree with this point. We have added literature

13. line 41: "A total of 1,154 mothers/guardians selected using three-stage cluster sampling procedure participated in the quantitative arm while purposively sampled one government senior 43 personnel, two paediatricians had KIIs and two FGDs with 9 mothers/guardians each." - Sentences could be made clearer, please improve grammar and provide more detail on the sampling procedure i.e. the 3 stages in the cluster sampling for the 1154 guardians, how the selection of each of government personnel, paediatricians and mothers/guardians for the qualitative component was done and specify what KII stands for.

Thank you. This has been corrected

14. line 50: "Of the 41 with recent U5C diarrhoea in the quantitative study, 11 obtained.." - should this not read "Of the 41 with recent U5C diarrhoea who used ORS/zinc in the quantitative study, 11 obtained.."?

Thank you. This has been corrected

15. line 51 - what is PMS the abbreviation for - please put the full words at first mention and the abbreviation in brackets.

Thank you. This has been corrected

16. line 51: improve grammar for readability: perhaps add from i.e. "6 each from a government hospital and a government health centre"

Thank you. This has been corrected

17. general: the finding that ORS/zinc is in fact available and free of charge at hospitals but many mothers reported unavailability should be emphasized and the reasons why mothers reported unavailability should be emphasized and discussed. lines 57-63 touch on this a little.

We agree with this point. We have added a line of discussion

18. general: of a population based study, the sample of 71 mothers reporting recent diarrhoea is quite small. The authors should comment on the relatively small sample size which is perhaps underpowered for assessing barriers to ORS. For example, was this sample of 1154 part of a bigger study that aimed to do.... and this study conducted a secondary analysis of the sample of 71 to assess patterns of barriers to ORS.

We thought as much. The original sample size was 556. The analysis of the 71 with recent episode wasn't secondary analysis but a cascaded part of the primary analysis. The 71/1154 shows that the prevalence is 6%. We are limited by word count to explain all these in this letter.

Reviewer: 2

Dr. Akanni Ibukun, Obafemi Awolowo University

19. line 44 "Oly" should read only.

Thank you. This has been corrected

20. How valid is the view of key informants from the hospital on availability of the product? Maybe, there are other ways either through mystery clients or physical confirmation of the product.

Thank you. These suggestions are very useful and will be adopted in further research

21. Promoting awareness and BCC as recommended is very appropriate. However, what channels will be more suitable to the targeted groups?

Thank you. We recommend community level awareness

VERSION 2 – AUTHOR RESPONSE

1. Title amend to "Barriers to utilization of oral rehydration solution and zinc in managing diarrhoea among under-five children in Oyo State, Nigeria"

Thank you. We have changed the title to read "Barriers to utilization of oral rehydration solution and zinc in managing diarrhoea among under-five children in Oyo State, Nigeria"